# Use of GLP1 receptor agonists in early pregnancy and reproductive safety: a multicentre, observational, prospective cohort study based on the databases of six Teratology Information Services

Kim Dao,[1] Svetlana Shechtman,[2] Corinna Weber-Schoendorfer,[3] Orna Diav-Citrin,[2,4] Reem Hegla Murad,[2] Maya Berlin ,[5] Ariela Hazan,[5] Jonathan L Richardson,[6] Georgios Eleftheriou,[7] Valentin Rousson,[8] Leonore Diezi,[1] David Haefliger,[1] Ana Paula Simões-Wüst ,[9] Marie-Claude Addor,[10] David Baud ,[11] Faiza Lamine,[12,13] Alice Panchaud,[14,15] Thierry Buclin,[1] François R Girardin,[1] Ursula Winterfeld [1]

FRG and UW contributed equally.

For numbered affiliations see end of article.

**Correspondence to**
Dr Ursula Winterfeld;
ursula.winterfeld@chuv.ch

## ABSTRACT

**Objectives** Glucagon-like peptide 1 receptor agonists (GLP1-RA) are indicated for the treatment of type 2 diabetes and more recently for weight loss. The aim of this study was to assess the risks associated with GLP1-RA exposure during early pregnancy.

**Design** This multicentre, observational prospective cohort study compared pregnancy outcomes in women exposed to GLP1-RA in early pregnancy either for diabetes or obesity treatment with those in two reference groups: (1) women with diabetes exposed to at least one non-GLP1-RA antidiabetic drug during the first trimester and (2) a reference group of overweight/obese women without diabetes, between 2009 and 2022.

**Setting** Data were collected from the databases of six Teratology Information Services.

**Participants** This study included 168 pregnancies of women exposed to GLP1-RA during the first trimester, alongside a reference group of 156 pregnancies of women with diabetes and 163 pregnancies of overweight/obese women.

**Results** Exposure to GLP1-RA in the first trimester was not associated with a risk of major birth defects when compared with diabetes (2.6% vs 2.3%; adjusted OR, 0.98 (95% CI, 0.16 to 5.82)) or to overweight/obese (2.6% vs 3.9%; adjusted OR 0.54 (0.11 to 2.75)). For the GLP1-RA group, cumulative incidence for live births, pregnancy losses and pregnancy terminations was 59%, 23% and 18%, respectively. In the diabetes reference group, corresponding estimates were 69%, 26% and 6%, while in the overweight/obese reference group, they were 63%, 29% and 8%, respectively. Cox proportional cause-specific hazard models indicated no increased risk of pregnancy losses in the GLP1-RA versus the diabetes and the overweight/obese reference groups, in both crude and adjusted analyses.

**Conclusions** This study offers reassurance in cases of inadvertent exposure to GLP1-RA during the first trimester of pregnancy. Due to the limited sample size, larger studies are required to validate these findings.

## STRENGTHS AND LIMITATIONS OF THIS STUDY

⇒ This observational prospective multicentre study aimed to assess the reproductive safety of early pregnancy exposure to glucagon-like peptide 1 receptor agonists (GLP1-RA), providing valuable insights into a previously underexplored area.

⇒ The study design incorporated two reference groups (diabetes and overweight/obese), to reduce the influence of potential confounding variables.

⇒ GLP1-RA were analysed as a single homogeneous group, only allowing for the exploration of potential class effects as specific GLP1-RA had limited individual exposure.

⇒ The availability of additional data, such as glycated haemoglobin or fasting glucose levels, varied within the study sample, which affected our ability to precisely describe disease severity among patients with diabetes.

## INTRODUCTION

Glucagon-like peptide 1 receptor agonists (GLP1-RA) have been widely used as therapeutic agents for the management of type 2 diabetes. More recently, certain GLP1-RA (liraglutide and semaglutide) have also gained approval in some countries for the treatment of obesity. The demand for GLP1-RA approved for obesity is high so there have even been supply shortages in Europe.[1] Since the prevalence of both overweight and obesity among women of childbearing age is increasing substantially in most countries (between 30% in some European countries and nearly 50% in the USA), it is to be expected that a further growing number of women of reproductive age will be treated

with GLP1-RA.[2–4] Approximately 50% of pregnancies worldwide are unplanned[5]; therefore, the question of the safety of GLP1-RA is particularly relevant.

This drug class includes short-acting GLP1-RA exenatide, lixisenatide and beinaglutide, as well as long-acting GLP1-RA dulaglutide, semaglutide, liraglutide and albiglutide (discontinued in 2017 at the request of the marketing authorisation holder due to steady decline in sales). By mimicking the actions of the hormone GLP1, these agents improve blood glycaemic control by increasing glucose-dependent insulin secretion, decreasing inappropriate glucagon secretion, and regulate appetite (by increasing satiety and delaying gastric emptying) to promote weight loss. The safety profile of GLP1-RA use during the first trimester of pregnancy has been investigated in previous studies, which did not reveal an elevated risk of major birth defects. One single case of exposure to liraglutide in the first trimester has been reported, describing a favourable outcome for the newborn.[6] A registry for exenatide documented seven cases of exposure during pregnancy, but follow-up information is lacking.[7] A case report describes the outcomes of two separate pregnancies in a 40-year-old woman with diabetes and obesity undergoing exenatide treatment. The report details the birth of one healthy child and another pregnancy resulting in a child with an atrial septal defect, which spontaneously resolved by the age of 3 years.[8] One single case of exposure to semaglutide in early pregnancy due to off-label treatment for polycystic ovary syndrome (PCOS) was reported with no birth defect.[9] Another case of exposure to dulaglutide during the first trimester of pregnancy for the treatment of type 2 diabetes did not result in any reported birth defects.[10] Due to the paucity of data concerning semaglutide and human pregnancy outcomes, the manufacturer advises stopping this treatment 2 months before conception.[11] In a multinational population-based cohort study involving 51,826 pregnant women diagnosed with type 2 diabetes and their infants, the standardised prevalence of major congenital malformations was 8.3% among infants with periconceptional exposure to GLP1-RA (n=938).[12] Compared with insulin, there was no increased risk of major congenital malformations (adjusted relative risk 0.95, 95% CI, 0.72 to 1.26) for infants exposed to GLP1-RA.

According to product labelling, animal studies have indicated a potential for reproductive toxicity at maternally toxic doses for semaglutide, dulaglutide, exenatide and liraglutide. An increased risk of malformations was observed for liraglutide and semaglutide at doses comparable to those administered in humans.[13–17] GLP1-RA are characterised by their high molar mass, ranging from 3700 g/mol (liraglutide) to 4100 g/mol (semaglutide and exenatide) and even up to 63 000 g/mol (dulaglutide). Even though additional physicochemical properties play important roles, placental transfer of drugs with such high molecular size is generally not anticipated in the first trimester of pregnancy, unless a specific mechanism for transfer exists.[18]

While previous studies did not detect an increased risk of major birth defects following the use of GLP1-RA during the first trimester of pregnancy, further confirmation from additional studies is warranted. The primary objective of this study was to prospectively evaluate the risk of major birth defects, spontaneous pregnancy losses (including abortions and stillbirths) and pregnancy terminations following first-trimester exposure to a GLP1-RA. Secondary objectives included describing additional pregnancy outcomes, such as gestational weeks at birth, birth weight and categorical neonatal outcomes like preterm birth and small or large for gestational age.

## MATERIAL AND METHODS
### Study design and dataset description
This multicentre, prospective, observational cohort study was conducted involving six participating centres that are members of the European Network of Teratology Information Services (ENTIS) in five countries: Germany, Israel, Italy, Switzerland, and the UK. ENTIS is an organisation of specialised services providing expertise on potential risks associated with medication exposure during pregnancy and breastfeeding at an individual level.[19] Standardised protocols are followed for data collection at each participating centre.[20] Methodological aspects pertaining to collaborative studies of a similar nature have been extensively documented in the existing literature, thus serving as a reference for performing the present study.[21]

Health professionals and pregnant women spontaneously contact a Teratology Information Service (TIS) for a risk assessment during pregnancy. Data collection is performed at this first contact with the TIS and following the anticipated date of delivery. Standardised questionnaires are used, administered to the patient and/or their healthcare provider. Maternal characteristics, such as age, tobacco use and alcohol consumption, as well as medical and obstetric history, are recorded. Detailed information regarding drug exposure and drug treatment indication, including dose, timing of therapy initiation, duration and concurrent medications, is also documented during the initial contact with the TIS. After the expected date of delivery, follow-up is conducted through structured mailed questionnaires and/or a structured telephone interview. The follow-up process seeks information on various aspects, including further medication use, pregnancy outcomes, gestational age at delivery, birth weight, presence of birth defects and neonatal complications. For this study, additional data pertaining to diabetes severity and obesity were also collected, where available. Socioeconomic and educational data were mostly not recorded.[22]

### Exposed and reference groups
The study sample was comprised of women exposed to any GLP1-RA and two reference groups: one of women with diabetes and the other of overweight/obese women without diabetes. The patients, or their healthcare providers, contacted one of the six participating TIS

between 2009 and 2022. Only patients with prospectively ascertained pregnancy outcomes, meaning those with unknown pregnancy outcomes or prenatal pathologies at the time of study enrolment, were included. The GLP1-RA exposed group consisted of pregnant women who used at least one GLP1-RA (identified by ATC codes A10BJ, A10AE54 or A10AE56) either as monotherapy or in combination with other medications during the first trimester of pregnancy.

The first reference group included pregnancies of patients diagnosed with pre-existing diabetes mellitus (International Statistical Classification of Diseases and Related Health Problems 10th Revision (ICD-10 codes E10–E13), who were exposed to non-GLP1-RA antidiabetic drugs during the first trimester of pregnancy (in most cases metformin). Patient selection for this reference group was done through a randomised process, choosing from the same prospective cohort within the corresponding TIS and within the same time frame of contact (±3 years). The second reference group primarily included pregnancies of patients classified as overweight (body mass index (BMI) ≥25 kg/m$^2$) or obese (BMI ≥30 kg/m$^2$) (ICD-10 code E66). They were aimed to be matched for BMI with the exposed group for a BMI of ±2 kg/m$^2$. Similarly, patient selection within this reference group was randomised from the same prospective cohort within the corresponding TIS and within the same time frame of contact (±3 years).

Exclusion criteria for all three groups included exposure to any teratogenic drugs such as systemic retinoids (including acitretin, alitretinoin, bexarotene, isotretinoin and tretinoin); cytotoxic agents and selected antiepileptic drugs (including valproate, carbamazepine, phenytoin, fosphenytoin, primidone, topiramate and phenobarbital); thalidomide, leflunomide, lenalidomide and coumarin derivatives (including dicoumarol, phenindione, warfarin, phenprocoumon, acenocoumarol and fluindione); and lithium, misoprostol, carbimazole and methimazole/thiamazole at any time during pregnancy. Additionally, exclusion criteria encompassed exposure to angiotensin-converting enzyme inhibitors, angiotensin II receptor blockers or tetracyclines during the second and third trimesters, presence of malignancies or malignancy-related conditions, multiple pregnancies and duplicate cases. Data sent from different TIS were analysed anonymously.[22]

## Outcomes

Birth defects were classified by two independent coauthors (MCA, DB) who were blinded to exposure data. The classification was performed using the European Network of Population-based Registries for the Epidemiological Surveillance of Congenital Anomalies (EUROCAT) ICD10-British Paediatric Association system.[23] The assessment of major birth defects was restricted to live births and pregnancy losses with confirmed outcomes after appropriate medical examination, ensuring reliable knowledge of birth defect status. Birth defects with genetic or chromosomal anomalies and those occurring due to intrauterine infections were excluded. Due to significant likelihood of under-reporting, minor birth defects were not included in the evaluation. Primary outcomes also included the risks of spontaneous pregnancy losses, which comprised both abortions and stillbirths as a combined outcome, categorised based on gestational age as either <22 weeks or ≥22 weeks, respectively. Additionally, the risks of pregnancy terminations were analysed. Secondary outcomes were preterm birth (<37 weeks of gestation after last menstrual period) and gestational age at birth and birth weight. Large (LGA) and small (SGA) for gestational age were also compared between cohorts. LGA was defined as a birth weight exceeding the 90th percentile, and SGA as birth weight less than the 10th percentile based on WHO infant sex-specific growth charts.[24]

## Statistical analysis

Crude risks of major birth defects were determined by dividing the total number of infants or fetuses with birth defects by the sum of all live-born infants, plus the number of cases with known birth defects in stillbirths and terminated pregnancies.[19] To assess the association between GLP1-RA exposure and the risk of major birth defects, multivariate logistic regression analysis was conducted, generating an OR with a corresponding 95% CI and adjusted for confounding factors, including maternal age (≤35 and >35 years), number of previous pregnancies and polymedication.[19] The number of previous pregnancies was categorised (0, 1, ≥2), and polymedication was defined as the use of more than one drug, including any other medication. Adjustment for potential confounding factors was done for covariates that were imbalanced between groups. An additional category for missing values was included to account for instances where data on any of the confounding factors, such as maternal age, history of pregnancies and polymedication, were not available.[7] In addition to the primary analysis, we conducted a sensitivity analysis to include all reported birth defects, irrespective of aetiology including those of genetic or chromosomal origin and those occurring due to intrauterine infections.

Elective pregnancy terminations for personal reasons (ETOP) and elective terminations for medical reasons, as well as spontaneous pregnancy losses (miscarriages and stillbirths), were treated as competing events in this study. The frequency of these outcomes was represented using cumulative incidence functions,[25] while delayed entries were treated as described by Rousson et al.[26] Women were thus considered to be at risk of experiencing one of these outcomes solely from their gestational age at the time of contact with the TIS. This allowed us to take into account the gestational age at the time of TIS contact, acknowledging that it was not independent of the outcome, given that certain pregnancy terminations may not be feasible at an advanced gestational age. Cases with missing information on gestational age at call or pregnancy outcome were excluded from the cumulative incidence analysis. To

assess the association between exposure and the overall risk of pregnancy loss, cause-specific Cox proportional hazard models were used, considering any variation in gestational age at enrolment across the three groups. In the hazard models, adjustment was performed for maternal age, squared maternal age and binary variables including tobacco use, folate supplementation, past ETOP and past abortions. Missing values for folate substitution were categorised as a separate category. Maternal ages for the missing values (n=8) were imputed using the median age of 34 years. Missing data on tobacco consumption (n=16), past ETOP (n=9) and past abortions (n=9) were imputed as 'no'.

Statistical analyses were conducted using R (V.4.3.1) and STATA V.17 (Stata Corp, College Station, Texas).[19 25]

The study protocol received approval from the ENTIS scientific committee. In most participating centres, this observational cohort study did not require ethics committee approval. However, in centres where it was necessary, appropriate ethics committee approval was obtained from the relevant authorities.

## Patient and public involvement
Patients and/or the public were not involved in the design, or conduct, or reporting, or dissemination plans of this research.

## RESULTS
In this study, we included 168 pregnant women who were exposed to GLP1-RA during the first trimester of pregnancy. Additionally, 156 pregnant women with a diagnosis of diabetes mellitus and 163 pregnant overweight or obese were included in two distinct reference groups.

## Maternal characteristics
Table 1 provides a summary of maternal characteristics, including obstetric and medical conditions. Women of the overweight/obese group were the youngest (median age 32 years) followed by those of GLP1-RA group (median age 34 years). The initial TIS contact of the group with diabetes was later than the other two groups. In comparison, the GLP1-RA group had the fewest primigravid women, with the majority of women having experienced more than two previous pregnancies.

In the group exposed to GLP1-RA, a substantial proportion of patients had a BMI of $25\,kg/m^2$ or higher (86.6%), and 27.4% had pregestational diabetes. Of the women with diabetes, 81.1% were overweight or obese. Medical conditions are summarised in table 1. Hypertension and dyslipidaemia were more frequently reported in the GLP1-RA exposed and in the reference group with diabetes. Psychiatric conditions were more frequently reported in the overweight/obese reference group.

## Drug exposure
Liraglutide (n=99) was the most commonly prescribed GLP1-RA followed by semaglutide (n=51), dulaglutide

(n=11) and exenatide (n=7). The indications for the use of GLP1-RA in the study included weight loss (n=117, 70%), diabetes (n=46, 28%) and other indications (n=4, 2%) such as PCOS, dumping syndrome or metabolic syndrome (online supplemental figure 1). In 2022, there was a remarkable surge in the number of women receiving GLP1-RA. Therapy was initiated prior to conception in 88% of the women in the exposed group and stopped at a median gestational age of 5 weeks (IQR 4–6 weeks; minimum, 2 weeks, maximum, 40 weeks). In the GLP1-RA exposed group, concomitant or subsequent use of anti-diabetic medications was reported in 44 (26.2%) cases, meaning in almost all women with diabetes (n=46) and of these, the use of more than one (non-GLP1-RA) antidiabetic medication was reported in 19 cases. Antidiabetic medications included insulin (n=21), metformin (n=19), sodium-glucose cotransporter 2 (SGLT2) inhibitors (n=5) and dipeptidyl peptidase 4 (DPP-4) inhibitors (n=1). In the reference group with diabetes (n=156), all women were treated with antidiabetic medication. The use of more than one antidiabetic medication was reported in 73 (46.8%) cases. It included metformin (n=139), insulin (n=63), DPP-4 inhibitors (n=23), sulfonylureas (n=12), SGLT2 inhibitors (n=6), thiazolidinediones (n=2) and repaglinide (n=3). In the overweight/obese reference group, the use of insulin was reported for the treatment of gestational diabetes in two cases (1%). A higher proportion of patients in both the diabetes and the overweight/obese reference groups was exposed to multiple medications (indicating the use of any medication).

## Pregnancy outcomes
Table 2 presents the proportions of offspring with major birth defects in pregnancies exposed to GLP1-RA agonists, the reference group with diabetes and the overweight/obese reference group. The rates of major birth defects, excluding genetic or chromosomal anomalies and those associated with intrauterine infections, were similar in both the GLP1-RA exposed group and the reference group with diabetes (2.6% and 2.3%, respectively). After adjustment for maternal age, the number of previous pregnancies and the use of more than one medication, the OR was 0.98 (95% CI, 0.16 to 5.82). In comparison with the overweight/obese reference group, which had a rate of major birth defects of 3.9%, the adjusted OR was 0.54 (95% CI, 0.11 to 2.75). In the GLP1-RA group, three major birth defects comprising one congenital heart defect, one congenital anomaly of the kidney and one case of multiple anomalies were observed, suggesting that these anomalies appear to be aetiologically distinct. Online supplemental table 1 provides comprehensive information on reported anomalies, concomitant medications and maternal conditions. In the sensitivity analysis, the rates of major birth defects, when all major anomalies irrespective of aetiology were included, were 3.1% in the GLP1-RA exposed group and 4.2% in the reference group with diabetes. The crude OR between the GLP1-RA exposed group and the reference group with diabetes was

**Table 1** Maternal/pregnancy characteristics in study groups

| Characteristics | GLP1-RA group (n=168) | Reference group with diabetes (n=156) | Overweight/obese reference group (n=163) |
|---|---|---|---|
| Maternal age (years), n | 166 | 156 | 158 |
| ≤35, n (%) | 102 (61.5) | 72 (46.2) | 107 (67.7) |
| >35, n (%) | 64 (38.6) | 84 (53.9) | 51 (32.3) |
| Tobacco use, n | 162 | 147 | 162 |
| Yes, n (%) | 21 (13.0) | 18 (12.2) | 16 (9.9) |
| Alcohol consumption, n | 162 | 138 | 162 |
| Yes, n (%) | 3 (1.9) | 2 (1.5) | 8 (4.9) |
| GA initial contact (weeks), n | 168 | 156 | 163 |
| Median (IQR) | 6 (5-9) | 7 (6-12) | 6 (5–9) |
| Previous pregnancies, n | 167 | 152 | 162 |
| 0, n (%) | 18 (10.8) | 39 (25.7) | 52 (32.1) |
| 1, n (%) | 31 (18.6) | 36 (23.7) | 42 (25.9) |
| ≥2, n (%) | 118 (70.7) | 77 (50.7) | 68 (42.0) |
| Previous spontaneous abortions, n | 167 | 151 | 160 |
| 0, n (%) | 120 (71.9) | 110 (72.9) | 121 (75.6) |
| 1, n (%) | 30 (18.0) | 26 (17.2) | 24 (15.0) |
| ≥2, n (%) | 17 (10.2) | 15 (9.9) | 15 (9.4) |
| Previous ETOP, n | 167 | 151 | 160 |
| 0, n (%) | 154 (92.2) | 134 (88.7) | 145 (90.6) |
| ≥1, n (%) | 13 (7.8) | 17 (11.3) | 15 (9.4) |
| Pregestational diabetes, n | 168 | 156 | – |
| n (%) | 46 (27.4) | 156 (100.0)* | –† |
| Overweight/obesity, n | 164 | 154 | 161 |
| BMI 25–30 kg/m$^2$, n (%) | 45 (27.4) | 37 (24.0) | 56 (34.8) |
| BMI >30 kg/m$^2$, n (%) | 97 (59.2) | 88 (57.1) | 103 (64.0) |
| Other medical conditions, n | 80 | 147 | 141 |
| Hypertension n (%) | 13 (16.3) | 31 (21.1) | 7 (5.0) |
| Dyslipidaemia n (%) | 7 (8.8) | 14 (9.5) | 0 (0) |
| Hypothyroidism n (%) | 8 (10.0) | 14 (9.5) | 16 (11.3) |
| Psychiatric condition n (%) | 9 (11.3) | 20 (13.6) | 52 (36.9) |
| Epilepsy n (%) | 1 (1.3) | 1 (0.7) | 0 (0) |
| Autoimmune disease n (%) | 5 (6.3) | 7 (4.8) | 14 (9.9) |
| Other n (%) | 24 (30.0) | 36 (24.5) | 54 (38.3) |
| Any medication in pregnancy, n | 168 | 156 | 163 |
| n (%) | 102 (60.7) | 139 (89.1) | 121 (74.2) |

*Inclusion criterion.
†Exclusion criterion.
BMI, body mass index; ETOP, elective termination of pregnancy; GA, gestational age; GLP1-RA, glucagon-like peptide 1 receptor agonists; IQR, Interquartile range; n, number of patients.

1.37 (95% CI, 0.36 to 5.23), and the adjusted OR was 0.99 (95% CI, 0.22 to 4.42). Compared with the overweight/obese reference group, which had a rate of major birth defects irrespective of aetiology of 6.3%, the crude OR was 0.66 (95% CI, 0.21 to 2.07), and the adjusted OR was 0.60 (95% CI, 0.16 to 2.21).

A total of 483 women were included in the cumulative incidence analysis, including 167 in the GLP1-RA group, 154 in the diabetes reference group and 162 in the overweight/obese reference group. In the GLP1-RA group, cumulative incidence estimates for live births, pregnancy losses and pregnancy terminations were 59%, 23% and

**Table 2** Risk of birth defects in pregnancies exposed to GLP1-RA during the first trimester compared with the two reference groups

| | GLP1-RA group | Reference group with diabetes | Overweight/obese reference group | OR (95% CI) GLP1-RA group vs reference group with diabetes | OR$_{adj}$ (95% CI)* GLP1-RA group vs reference group | OR (95% CI) GLP1-RA group vs overweight/ obese reference group | OR$_{adj}$ (95% CI)* |
|---|---|---|---|---|---|---|---|
| Major birth defects excluding chromosomal/genetic/infection associated, n (%) | 3/117 (2.6)† | 3/128 (2.3)‡ | 5/127 (3.9)§ | 1.10 (0.22 to 5.54) | 0.98 (0.16 to 5.82) | 0.64 (0.15 to 2.75) | 0.54 (0.11 to 2.75) |
| Major birth defects—subgroups of anomalies | | | | | | | |
| Congenital anomalies of kidney, n | 1 | | | | | | |
| Urogenital anomalies, n | | | 1 | | | | |
| Congenital heart defects, n | 1 | | 2 | | | | |
| Nervous system anomalies, n | | 1 | 1 | | | | |
| Polymalformative syndrome, n | 1 | 1 | | | | | |
| Limb anomalies, n | | 1 | | | | | |
| Gastrointestinal anomalies, n | | | 1 | | | | |

*Adjusted for maternal age (≤35 years and >35 years), number of previous pregnancies (0, 1, ≥2), use of more than one medication (yes/no).
†Limited to pregnancies with available data on the presence or absence of a birth defect, including 113 live births, 3 terminated pregnancies and 1 stillbirth.
‡Limited to pregnancies with available data on the presence or absence of a birth defect, including 126 live births, 1 terminated pregnancy and 1 stillbirth.
§Limited to pregnancies with available data on the presence or absence of a birth defect, including 117 live births, 7 terminated pregnancies and 3 stillbirths.
GLP1-RA, glucagon-like peptide 1 receptor agonists.

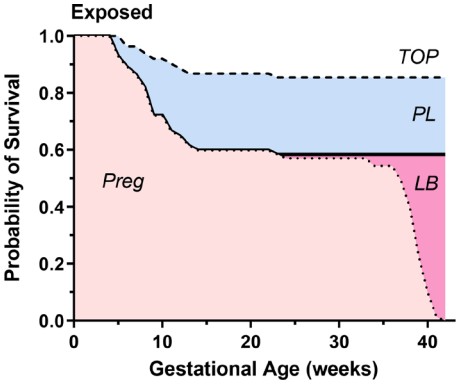

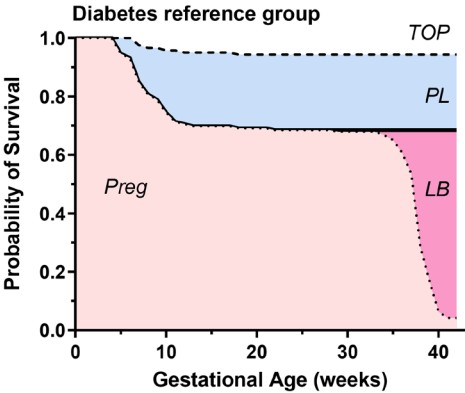

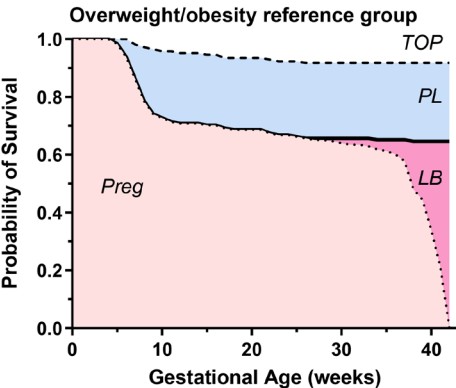

**Figure 1** Cumulative incidences of pregnancy outcomes with live birth (LB), pregnancy losses (PL) and pregnancy termination (TOP) in women exposed to glucagon-like peptide 1 receptor agonists, the diabetes reference group and the overweight/obesity reference group.

18%, respectively (figure 1). The corresponding estimates in the diabetes reference group were 69%, 26% and 6%, while in the overweight/obese reference group, they were 63%, 29% and 8%.

Details for other pregnancy and neonatal outcomes are presented in table 3. The rate of preterm births was almost doubled in the group with diabetes (15.1%) and in the overweight/obese group (14.5%) compared with the GLP1-RA group (8.0%). The rates of infants classified as LGA were higher in the GLP1-RA exposed group and in the diabetes reference group compared with the overweight/obese reference group. The Cox proportional

cause-specific hazard models (table 4) revealed no increased risk of pregnancy loss in either the comparison between the GLP1-RA group and the diabetes reference group or the overweight/obese reference group. However, in the unadjusted and adjusted analysis, the GLP1-RA group showed an increased risk of termination of pregnancy (TOP) compared with the diabetes reference group (HR, 2.92; 95% CI, 1.18 to 7.20; p=0.02; adjusted HR (HRadj), 3.89; 95% CI, 1.48 to 10.2; p=0.01). Nonetheless, no significant increase in the risk of TOP was observed when the GLP1-RA group was compared with the overweight/obese reference group (HR, 1.79; 95% CI, 0.89 to 3.62; p=0.10; HRadj, 1.39; 95% CI, 0.66 to 2.93; p=0.38.

### Follow-up

Information on follow-up was obtained from healthcare professionals and patients in similar proportions for both the GLP1-RA and the reference groups with diabetes and who were overweight/obese: healthcare professionals 10% vs 9% and 6% and patients 90% vs 91% and 94%, respectively. Infant age at follow-up was reported for 61% in the GLP1-RA group, 73% in the diabetes group and 72% in the overweight/obese group. The infant age at follow-up was similar in the GLP1-RA group (neonates aged 0–28 days, 8%; infants aged 28 days to 2 years, 76%; and children aged >2 years, 16%), the diabetes reference group (neonates, 0%; infants, 63%; and children, 37%) and the overweight/obese reference group (neonates, 4%; infants, 87%; and children, 9%).

### DISCUSSION

This prospective multicentre observational study adds further evidence by assessing pregnancy outcomes following exposure in early pregnancy to GLP1-RA. Examining 168 pregnant women exposed to a GLP1-RA during the first trimester of pregnancy, alongside two reference groups (comprising pregnant women diagnosed with diabetes mellitus and overweight/obese pregnant women), we did not identify a specific pattern of birth defects. Furthermore, our analysis revealed no association between GLP1-RA exposure and an increased risk of major birth defects. These findings align with a recently published multinational population-based cohort study. In comparison with insulin, this study also found no increased risk of major congenital malformations for infants exposed to GLP1-RA during the periconceptional period.[12]

The crude rate of major birth defects in the GLP1-RA-exposed women was 2.6%, aligning with the prevalence of major birth defects as reported by EUROCAT which was 2.6% including genetic anomalies for the years 2005–2021.[27–29] This rate is also equivalent to that observed in the reference group with diabetes in our study (2.3%). As diabetes with poor glycaemic control is associated with an increased risk of major birth defects,[30] the rate of 2.3% seems low. Indeed, the rate of major birth defects often

**Table 3** Pregnancy outcomes

| | GLP1-RA group (n=168) | Reference group with diabetes (n=156) | Overweight/obese reference group (n=163) |
|---|---|---|---|
| Live-born infants, n | 113 | 126 | 117 |
| Elective termination of pregnancy for personal reasons, n | 20 | 5 | 6 |
| Medical termination of pregnancy, n | 3* | 1† | 7‡§ |
| Spontaneous abortion, n | 31 | 23 | 29 |
| Stillbirth, n | 1 | 1 | 4 |
| Preterm delivery, n (%) (n=113, 126, 127) | 9 (8.0) | 19 (15.1) | 17 (14.5) |
| Gestational age at birth (week), median (IQR) (n=113, 125, 117) | 39 (38–40) | 38 (37–39) | 39 (38–40) |
| Birth weight (g), median (IQR) (n=113, 124, 117) | 3400 (3100–3660) | 3370 (2890–3682) | 3245 (2850–3560) |
| Large for gestational age, n (%) (n=113, 124, 117) | 20 (17.7) | 29 (23.4) | 7 (6.0) |
| Small for gestational age, n (%) (n=113, 124, 117) | 9 (8.0) | 10 (8.1) | 17 (14.5) |
| Delivery mode, n (%) (n=112, 135, 108) | | | |
| Spontaneous vaginal delivery | 60 (53.6) | 68 (50.4) | 60 (55.6) |
| Assisted delivery and caesarean section | 52 (46.4) | 67 (49.6) | 48 (44.4) |

*Reason for medical termination of pregnancy: one congenital anomaly, one genetic anomaly and one caesarean scar pregnancy.
†Reason for medical termination of pregnancy: one congenital anomaly.
‡Reason for medical termination of pregnancy: five congenital anomalies (see table 2); in addition, one was due to an intrauterine cytomegalovirus infection associated with anomalies, one genetic anomaly.
§One ectopic pregnancy.
GLP1-RA, glucagon-like peptide 1 receptor agonists; IQR, Interquartile range.

lies between 5% and 10% among women with pregestational diabetes[25 31–33] but depends on the maternal blood sugar level. There is a linear positive correlation between the risk of major congenital anomalies and pre-existing diabetes. The risk of congenital malformations in offspring of mothers with diabetes generally exceeds the risk in the general population as soon as periconception glycated haemoglobin (HbA1C) exceeds 6.5%.[34] Unfortunately, we lacked systematic access to glycated haemoglobin values, which could have indicated that cases in our study population were with well-managed diabetes inadvertently selected leading to a rate of birth defects similar to the general population. In addition, the low risk of TOPs in the reference group with diabetes might reflect a higher proportion of planned pregnancies, benefiting from better diabetes control. We did not observe an

increased risk of pregnancy losses when comparing the GLP1-RA group with the two reference groups. However, the higher incidence of elective terminations for personal reasons in the GLP1-RA group, compared with both reference groups, may be indicative of both a greater number of unplanned pregnancies and anxiety related to the unknown risks of GLP1-RA medication for the fetus.

Interestingly, 70% of the exposed women took the drug for weight reduction, although the primary drug treatment indication remains diabetes type 2. As of 2018, an increasing number of women have used GLP1-RA for weight loss (see online supplemental figure 1).[4]

Strengths and limitations of prospective observational pregnancy cohort studies based on ENTIS data have already been described in the literature.[20] The study design with two reference groups (diabetes and

**Table 4** Pregnancy outcomes—Cox proportional hazard models

| | GLP1-RA group vs diabetic reference group | | GLP1-RA group vs overweight/obese reference group | |
|---|---|---|---|---|
| | HR (95% CI) | $HR_{adj}$ (95% CI)* | HR (95% CI) | $HR_{adj}$ (95% CI)* |
| Pregnancy loss† | 1.10 (0.65 to 1.89) | 1.67 (0.93 to 3.01) | 0.92 (0.56 to 1.49) | 0.80 (0.48 to 1.33) |
| Pregnancy termination‡ | 2.92 (1.18 to 7.20) | 3.89 (1.48 to 10.2) | 1.79 (0.89 to 3.62) | 1.39 (0.66 to 2.93) |

*Adjusted for maternal age, tobacco use, folate intake, past abortion or termination history.
†Spontaneous abortions and stillbirths.
‡Elective or medical termination of pregnancy.
GLP1-RA, glucagon-like peptide 1 receptor agonists; $HR_{adj}$, adjusted HR.

overweight/obese) served to mitigate the influence of potential confounding factors. However, additional data (in particular glycated haemoglobin) would have enabled a more precise description of the severity of the disease in patients with diabetes. Given the primary focus of our study on exposures during the first trimester, our analysis did not consider the potential risks associated with gestational diabetes. Furthermore, the treatment with GLP1-RA was discontinued during early gestational ages, in most cases. Thus, regarding cases with exposure to GLP1-RA of shorter half-lives, such as liraglutide and exenatide, the exposure window did not encompass the entire first trimester of pregnancy in most cases. GLP1-RA were analysed as a homogeneous group, to explore a potential class effect. When taken individually, only a limited number of women were exposed to specific GLP1-RA such as dulaglutide and exenatide, and there were no cases of exposure to albiglutide and beinaglutide. Finally, the sample size is not large enough to draw firm and definitive conclusions.

## CONCLUSION

This study offers further reassurance in cases of inadvertent exposure to GLP1-RA during the first trimester of pregnancy. Documentation and follow-up of these pregnancies are important to allow for further studies on a broader scale.

**Author affiliations**
[1]Swiss Teratogen Information Service and Clinical Pharmacology Service, Centre Hospitalier Universitaire Vaudois (CHUV) and University of Lausanne, Lausanne, Switzerland
[2]The Israeli Teratology Information Service, Ministry of Health, Jerusalem, Israel
[3]Charité - Universitätsmedizin Berlin, Pharmakovigilanzzentrum Embryonaltoxikologie, Institut für Klinische Pharmakologie und Toxikologie, Berlin, Germany
[4]The Hebrew University and Hadassah Medical School, Jerusalem, Israel
[5]Clinical Pharmacology and Toxicology Unit, Drug Consultation Center, Zerifin TIS, affiliated with the Faculty of Medicine, Tel Aviv University, Shamir Medical Center Assaf Harofeh, Tzrifin, Central, Israel
[6]The UK Teratology Information Service, Newcastle Upon Tyne Hospitals NHS Foundation Trust, Newcastle Upon Tyne, UK
[7]Poison Control Center, Hospital ASST Papa Giovanni XXIII, Bergamo, Italy
[8]Center for Primary Care and Public Health, University of Lausanne, Lausanne, Switzerland
[9]Department of Obstetrics, University Hospital Zurich, University of Zurich, Zurich, Switzerland
[10]Department of Woman-Mother-Child, Centre Hospitalier Universitaire Vaudois (CHUV), Lausanne, Switzerland
[11]Materno-Fetal and Obstetrics Research Unit, Department Woman-Mother-Child, Centre Hospitalier Universitaire Vaudois (CHUV) and University of Lausanne, Lausanne, Switzerland
[12]Endocrinology, Diabetes and Metabolism Service, Centre Hospitalier Universitaire Vaudois (CHUV), Lausanne, Switzerland
[13]Endocrinology and Diabetes Unit. Internal Medicine Service, Hôpital Riviera-Chablais, Rennaz, Switzerland
[14]Institute of Primary Health Care (BIHAM), University of Bern, Bern, Switzerland
[15]Service of Pharmacy, Centre Hospitalier Universitaire Vaudois (CHUV), Lausanne, Switzerland

**Acknowledgements** The authors would like to thank pharmacists Sarit Carmi and Shira Shalit, from the Division of Clinical Pharmacy, Institute for Drug Research, School of Pharmacy, Hebrew University and Hadassah Medical School, Jerusalem, Israel, for their valuable contribution to the data collection, done while acquiring their PharmD degree. We thank the whole team at Swiss Teratogen Information Service (STIS) for their support, in particular the computer scientist Mr F Veuve.

**Contributors** KD, SS, OD-C, AP, APS-W, UW contributed to the study conception and design. KD, SS, CW-S, MB, AH, JLR, GE, RHM, OD-C, UW contributed to the acquisition of the data. KD, CW-S, VR, LD, DH, M-CA, DB, AP, OD-C, TB, FRG, UW contributed to data analysis. KD, SS, CW-S, MB, AH, JLR, GE, RHM, VR, LD, DH, APS-W, M-CA, DB, AP, FL, OD-C, TB, FRG, UW contributed to the interpretation of data and drafting and revising the manuscript for intellectual content. The guarantor of the study is UW.

**Funding** This work was supported by Swiss National Science Foundation grant number SNSF 320030_214844.

**Competing interests** None declared.

**Patient and public involvement** Patients and/or the public were not involved in the design, or conduct, or reporting, or dissemination plans of this research.

**Patient consent for publication** Not applicable.

**Ethics approval** This study involves human participants and was approved by the Institutional Review Board of Charité - Universitätsmedizin Berlin (Ref: EA2/041/23), Ministry of Health Jerusalem (Ref: MOH-072-2023 and Ref: ASF-0036-23), Shamir Medical Center. In the other centres (STIS Lausanne, UK Teratology Information Service, Newcastle upon Tyne and Poison Control Center Bergamo), no ethics approval was needed.

**Provenance and peer review** Not commissioned; externally peer reviewed.

**Data availability statement** Data are available in a public, open access repository. Source data related to the cohort study are available on https://zenodo.org.

**ORCID iDs**
Maya Berlin http://orcid.org/0000-0003-4731-742X
Ana Paula Simões-Wüst http://orcid.org/0000-0002-4489-0952
David Baud http://orcid.org/0000-0001-9914-6496
Ursula Winterfeld http://orcid.org/0000-0002-9926-7164

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
