## [Reviewer comments · BMJ Open]

ARTICLE DETAILS

TITLE (PROVISIONAL)	Use of GLP1 receptor agonists in early pregnancy and reproductive safety: a multicentre, observational, prospective cohort study based on the databases of six Teratology Information Services
AUTHORS	Dao, Kim; Shechtman, Svetlana; Weber-Schoendorfer, Corinna; Diav-Citrin, Orna; Murad, Reem; Berlin, Maya; Hazan, Ariela; Richardson, Jonathan L; Eleftheriou, Georgios; Rousson, Valentin; Diezi, Leonore; Haefliger, David; Simões-Wüst, Ana Paula; Addor, Marie-Claude; Baud, David; Lamine, Faiza; Panchaud, Alice; Buclin, Thierry; Girardin, François; Winterfeld, Ursula

VERSION 1 – REVIEW

REVIEWER	Luo, Zhong-Cheng Lunenfeld-Tanenbaum Research Institute, Obstetrics and Gynecology, Mount Sinai Hospital
REVIEW RETURNED	21-Jan-2024

GENERAL COMMENTS	The authors present their data on the incidence rates of major birth defects in women exposed glucagon-like peptide 1 receptor agonists (GLP1-RA) in early pregnancy as compared to two reference groups: pregnancies with diabetes using other anti-diabetic medications, pregnancies in women who were overweight or obese. GLP1-RA is a common medication in the treatment of type 2 diabetes, and is often used for weight loss in more recent years. They observed comparable incidence rates of major birth defects in the CLP1-RA and the two reference groups. Adjusted analyses confirmed the lack of risk difference. In general, the study is well conducted. The lack of association is reassuring concerning the reproductive health safety of GLP1-RA in humans. Main comments In the analysis, the authors have excluded birth defects with genetic or chromosomal abnormalities and those related to infections. Although this seems somewhat reasonable, it is difficult to be sure that any medication in pregnancy did not play a role in any specific case of birth defect. I would suggest a sensitivity analysis to include all birth defects, and show whether you have similar findings. About the regression models (Table 2, Table 4), please clarify how you select the co-variables in the adjusted model in the footnotes to Tables, or in the Methods section. About maternal age, why use it as a continuous variable? Most often, age is used as a dichotomous variable in perinatal research, as the risk increase is
--

	not linear, and women at >35 years are considered the high-risk group for birth defects. Minor comments The caption of Table 1, should be “Maternal/pregnancy characteristics in study groups” The caption of Table 4, should replace “Cox analysis” to “Cox proportional hazards models“
--	--

REVIEWER	Evers, Katrina University Children’s Hospital Basel, Pediatric Nephrology
REVIEW RETURNED	14-Feb-2024

GENERAL COMMENTS	In this manuscript the authors assess the risks associated with GLP1-RA exposure during early pregnancy by analyzing databases from six Teratology Information Services in a multicentre, observational prospective cohort study. Minor comments: Page 14 Line 31: It is not clear why the cumulative incidence estimates for live births, pregnancy losses and pregnancy terminations do not sum up to 100%, where does the small rest percentage come from? Page 7 Line 19: the citation is a bit confusing - it is not clear, that the same woman was preg
---

VERSION 1 – AUTHOR RESPONSE

Reviewer: 1

Dr. Zhong-Cheng Luo, Lunenfeld-Tanenbaum Research Institute, Shanghai Jiaotong University School of Medicine Xinhua Hospital

Comments to the Author:

The authors present their data on the incidence rates of major birth defects in women exposed glucagon-like peptide 1 receptor agonists (GLP1-RA) in early pregnancy as compared to two reference groups: pregnancies with diabetes using other anti-diabetic medications, pregnancies in women who were overweight or obese. GLP1-RA is a common medication in the treatment of type 2 diabetes, and is often used for weight loss in more recent years. They observed comparable incidence rates of major birth defects in the GLP1-RA and the two reference groups. Adjusted analyses confirmed the lack of risk difference. In general, the study is well conducted. The lack of association is reassuring concerning the reproductive health safety of GLP1-RA in humans.

Main comments

In the analysis, the authors have excluded birth defects with genetic or chromosomal abnormalities and those related to infections. Although this seems somewhat reasonable, it is difficult to be sure that any medication in pregnancy did not play a role in any specific case of birth defect. I would suggest a sensitivity analysis to include all birth defects, and show whether you have similar findings.

2) Response: Thank you for your valuable feedback on our manuscript. In response to your suggestion, we conducted a sensitivity analysis to include all major birth defects, irrespective of

aetiology, to assess whether our findings remain consistent. Our findings suggest that the inclusion of all reported birth defects did not significantly alter the risk estimates observed in the primary analysis.

We added the following sentence to the methods section (page 9): In addition to the primary analysis, we conducted a sensitivity analysis to include all reported birth defects, irrespective of aetiology including those of genetic or chromosomal origin and those occurring due to intrauterine infections.

Furthermore, we added the following sentences to the results section (page 12): In the sensitivity analysis, the rates of major birth defects, when all major anomalies irrespective of aetiology were included, were 3.1% in the GLP1-RA exposed group and 4.2% in the diabetic reference group. The crude OR between the GLP1-RA exposed group and the diabetic reference group was 1.37 (95% CI, 0.36-5.23), and the adjusted OR was 0.99 (95% CI, 0.22-4.42). Compared to the overweight/obese reference group, which had a rate of major birth defects irrespective of aetiology of 6.3%, the crude OR was 0.66 (95% CI, 0.21-2.07) and the adjusted OR was 0.60 (95% CI, 0.16-2.21).

About the regression models (Table 2, Table 4), please clarify how you select the co-variables in the adjusted model in the footnotes to Tables, or in the Methods section. About maternal age, why use it as a continuous variable? Most often, age is used as a dichotomous variable in perinatal research, as the risk increase is not linear, and women at >35 years are considered the high-risk group for birth defects.

3) Response: Regarding the selection of covariates in the adjusted models (Table 2, Table 4), we assessed differences in maternal characteristics across study groups using statistical tests like the Kruskal-Wallis test for continuous variables and the chi-square test for categorical variables and included variables that showed statistically significant differences among groups as covariates in the adjusted models. We have added a clarification in the Methods section (page 9). We also acknowledge your suggestion regarding maternal age. In response, we have revised our approach and now treat maternal age as a dichotomous variable (≤ 35 years, > 35 years) rather than a continuous variable.

Minor comments

The caption of Table 1, should be “Maternal/pregnancy characteristics in study groups”

The caption of Table 4, should replace “Cox analysis” to “Cox proportional hazards models”

4) Response: we have implemented the proposed changes accordingly.

Reviewer: 2

Dr. Katrina Evers, University Children's Hospital Basel

Comments to the Author:

In this manuscript the authors assess the risks associated with GLP1-RA exposure during early pregnancy by analyzing databases from six Teratology Information Services in a multicentre, observational prospective cohort study.

Minor comments:

Page 14 Line 31: It is not clear why the cumulative incidence estimates for live births, pregnancy losses and pregnancy terminations do not sum up to 100%, where does the small rest percentage come from?

5) Response: We greatly appreciate the reviewer's insightful observation. Upon careful review, we identified an inaccuracy in our calculation, which has been rectified in the revised version of the article (abstract and page 12). Notably, the cumulative incidences now sum up to 100% across all groups. However, it is important to note that in the diabetic control group, the cumulative incidences sum up to 101% due to rounding errors.

Page 7 Line 19: the citation is a bit confusing - it is not clear, that the same woman was pregnant twice.

6) Response: In response to your suggestion, we changed our sentence as follows: A case report describes the outcomes of two separate pregnancies in a 40-year-old woman with diabetes and obesity undergoing exenatide treatment. The report details the birth of one healthy child and another pregnancy resulting in a child with an atrial septal defect, which spontaneously resolved by the age of three.

VERSION 2 – REVIEW

REVIEWER	Luo, Zhong-Cheng Lunenfeld-Tanenbaum Research Institute, Obstetrics and Gynecology, Mount Sinai Hospital
REVIEW RETURNED	28-Mar-2024

GENERAL COMMENTS	The authors have adequately addressed all comments, and the revised manuscript is much improved.
--

REVIEWER	Evers, Katrina University Children's Hospital Basel, Pediatric Nephrology
REVIEW RETURNED	21-Mar-2024

GENERAL COMMENTS	The Authors have addressed all of my concerns with the original manuscript. The revised manuscript is ready for publication
---